# The Enigma of Eosinophil Degranulation

**DOI:** 10.3390/ijms22137091

**Published:** 2021-06-30

**Authors:** Timothée Fettrelet, Lea Gigon, Alexander Karaulov, Shida Yousefi, Hans-Uwe Simon

**Affiliations:** 1Institute of Pharmacology, University of Bern, Inselspital, INO-F, CH-3010 Bern, Switzerland; timothee.fettrelet@unil.ch (T.F.); lea.gigon@pki.unibe.ch (L.G.); shida.yousefi@pki.unibe.ch (S.Y.); 2Department of Biochemistry, University of Lausanne, CH-1066 Epalinges, Switzerland; 3Department of Clinical Immunology and Allergology, Sechenov University, 119991 Moscow, Russia; drkaraulov@mail.ru; 4Laboratory of Molecular Immunology, Institute of Fundamental Medicine and Biology, Kazan Federal University, 420012 Kazan, Russia; 5Institute of Biochemistry, Medical School Brandenburg, D-16816 Neuruppin, Germany

**Keywords:** degranulation, eosinophils, eosinophil extracellular trap, granule proteins, piecemeal degranulation, exocytosis, cytolysis

## Abstract

Eosinophils are specialized white blood cells, which are involved in the pathology of diverse allergic and nonallergic inflammatory diseases. Eosinophils are traditionally known as cytotoxic effector cells but have been suggested to additionally play a role in immunomodulation and maintenance of homeostasis. The exact role of these granule-containing leukocytes in health and diseases is still a matter of debate. Degranulation is one of the key effector functions of eosinophils in response to diverse stimuli. The different degranulation patterns occurring in eosinophils (piecemeal degranulation, exocytosis and cytolysis) have been extensively studied in the last few years. However, the exact mechanism of the diverse degranulation types remains unknown and is still under investigation. In this review, we focus on recent findings and highlight the diversity of stimulation and methods used to evaluate eosinophil degranulation.

## 1. Introduction

Eosinophils are specialized white blood cells that have been traditionally perceived as cytotoxic effector cells and recently suggested to be involved in immunomodulatory and homeostatic functions [1,2]. Eosinophils are found in all vertebrate species where they appear largely evolutionarily conserved with regards to their morphology and general functions [2]. Particularly, human and mouse eosinophils share similarities in distribution throughout the body, developmental patterns, cytokine response and degranulation kinetics [3,4]. Nonetheless, human and mouse eosinophils display differences in cell surface receptors, granule components and degranulation after activation with specific stimuli [2].

The occurrence of granules in the cytoplasm defines the hallmark of granulocytes—a subtype of leukocytes comprising neutrophils, eosinophils and basophils [5]. The formation of the distinct eosinophilic granules occurs at different stages of eosinophil maturation. The larger, primary granules are found low in number and are formed during the progranulocyte stages [6]. Charcot–Leyden crystal protein (CLC, also known as galectin 10) and eosinophil peroxidase (EPX, also known as EPO) are found in the primary eosinophil granules [2]. In this context, it should be mentioned that primary granules were recently suggested to be early “specific” cored granules [7]. The specific granules, also termed crystalloid or secondary granules, are formed during the myelocyte stage and are characterized by their crystalline core [6]. Eosinophil specific granules contain four cationic proteins—major basic protein (MBP-1), EPX, eosinophil cationic protein (ECP) and eosinophil-derived neurotoxin (EDN)—as well as numerous pre-formed cytokines and chemokines [8,9,10]. Lipid bodies, a typical constituent in leukocytes, comprise different types of proteins, growth factors, cytokines and chemokines [11]. Lipid bodies are involved in regulation of eicosanoid synthesis, control of lipid metabolism, membrane trafficking and intracellular signaling, as well as in immunoregulatory functions [11]. Additionally, small granules encompassing various proteins are found in mature eosinophils. Eosinophils can be stimulated by several ligands of C-C chemokine receptor type 3 (CCR3) including eotaxins, which are a CC chemokine subfamily of eosinophil chemotactic proteins [12]. Eotaxin-1 regulates the recruitment of eosinophils into the thymus and is therefore constitutively expressed in the thymus [13]. Furthermore, eotaxins are also crucial for the migration of eosinophils into tissue at sites of inflammation.

Eosinophils are known to be actively involved in several infectious, allergic and autoimmune diseases. A key event in the pathology of many allergic and nonallergic inflammatory diseases is the process of eosinophil degranulation, which describes the release of granule content into the extracellular space following the stimulation of eosinophils [14]. To date, four major mechanisms are defined for the process of degranulation in eosinophils: piecemeal degranulation (PMD), cytolysis, classical and compound exocytosis [15] (Figure 1). PMD was first discovered in basophils [16] and further described in diverse inflammatory cells [17,18]. PMD is characterized by the progressive and selective release of granule content with the help of small vesicles in absence of fusion to other granules or plasma membrane [19,20]. In the process of cytolysis, a nonapoptotic rapid form of cell death, intact granules are released following the rupture of the plasma membrane [21,22,23]. Exocytosis is defined by the release of granule or vesicular content into the extracellular space resulting from the fusion of granule directly with the plasma membrane (classical exocytosis) or from intracellular granule–granule fusion prior to interaction with plasma membrane (compound exocytosis) [15,24]. Exocytosis has been reported in several immune cells including mast cells [25,26], eosinophils [27], neutrophils [28], platelets [29] and NK cells [30].

The exact role of eosinophils in health and disease continues to be a matter of debate. In this review article, we aimed to summarize new findings regarding the process of eosinophil degranulation.

## 2. Molecular Regulation of Eosinophil Degranulation

The patterns and ultrastructural characteristics of the different eosinophil secretion processes have been extensively reviewed in the last few years [2,15,31]. Here, we detail the known molecular pathways and regulators of eosinophil degranulation.

Rab27a is a GTP-binding protein that has previously been shown to be expressed in eosinophils of asthmatic donors [32]. Recently, Kim et al. observed a role of Rab27a in eosinophil degranulation. Rab27a was shown to colocalize with eosinophil specific granules at the cell membrane, as well as with CD63^+^ granules following platelet-activating factor (PAF) stimulation. Furthermore, the absence of Rab27a reduced ionomycin- and PAF-induced degranulation, as assessed by EPX release measurement [33].

The SNARE proteins constitute a protein superfamily involved in the fusion of membranes. Some studies have described the occurrence of SNARE proteins intracellularly in human eosinophils (Figure 2). The vesicle-associated membrane protein 2 (VAMP-2) is mainly expressed in secretory vesicles [34] and has been suggested to be involved in PMD by interacting with t-SNAREs syntaxin-4 (STX4) and SNAP23 [35]. VAMP-7 [36,37] and VAMP-8 [36] are mostly related to specific granules. These findings hit on a specific distribution of SNARE proteins within eosinophils and therefore suggest a specific role of the SNARE proteins in eosinophil degranulation processes [36,37]. Willetts et al. demonstrated that VAMP-7-deficient eosinophils release less EPX following PAF, ionomycin or interleukin-33 (IL-33) stimulation as well as less MBP-1 following PAF or ionomycin stimulation [37]. These results confirmed previous data showing an impaired release of EPX and EDN after VAMP-7 inhibition [36]. In contrast to the impaired granule protein release, most of the released cytokines and chemokines did not show a significant decrease following activation of VAMP-7-deficient eosinophils [37]. Consequently, it has been suggested that the SNARE protein VAMP-7 is mostly involved in the sorting, mobilization and the release of granule proteins and that the cytokines/chemokines relies on a different pathway [37]. It is important to note that eosinophils were still viable (≥90%) following the stimulation with multiple agonists [37]. Interestingly, a reduced airway hyperresponsiveness (AHR) was observed in VAMP-7-deficient eosinophils, suggesting that eosinophil degranulation is also important for asthma exacerbations [37]. This stands in contrast with a recent experimental study showing that the absence of MPB or EPX in eosinophils does not improve AHR [38]. Thus, it is necessary to further investigate the exact role of eosinophil granule proteins and eosinophil degranulation in asthma pathogenesis.

A few years ago, the ancient Qa-SNARE protein syntaxin-17 (STX17) was reported to be present on secretory granules and eosinophil sombrero vesicles (eoSVs) in eosinophils [39]. Following activation with CCL11, a known agonist of PMD, the authors demonstrated a significant increase in the number of STX17-labeled secretory granules compared to unstimulated cells as assessed by immunonanogold transmission electron microscopy (TEM). Conversely, this increase was not observed following tumor necrosis factor-α (TNF-α) stimulation [39], which is known to induce compound exocytosis. Moreover, STX17 was unlikely involved in constitutive secretion as endoplasmic reticulum (ER) cisternae, and the Golgi complex regions were not positively labelled for STX17 [39]. These findings suggest a specific role of STX17 in the PMD secretory pathway [39].

The cyclin-dependent kinase 5 (Cdk5) and its effector molecules p35 and p39 are known to play a key role in neuronal cell exocytosis through the phosphorylation of Munc18, a regulator of SNARE binding [40]. Odemuyiwa et al. have demonstrated that human eosinophils express Cdk5, Munc18c (a protein able to bind to STX4), p35 and p39. Furthermore, they observed a physical interaction between Cdk5 and the molecules p35 and p39 as well as an association of Cdk5 with Munc18c following eosinophil activation [41]. Cdk5 showed increased kinase activity in eosinophils following stimulation with CCL11 and PAF, both mostly recognized as agonists of PMD (Table 1). Moreover, broad Cdk-specific inhibitors and Cdk5-specific siRNA (partially knocking down Cdk5) reduced the release of EPX following phorbol 12-myristate 13-acetate (PMA) stimulation, suggesting that Cdk5 is a key element involved in the regulation of granule proteins exocytosis [41]. Unfortunately, the authors did not show data of EPX release from human eosinophils following stimulation with the physiological agonists CCL11 and PAF.

Autophagy and the altered expression of autophagy-related (ATG) proteins are known to play a role in the regulation of eosinophil functions [42,43]. *ATG5*-knockout eosinophils exhibited an increase in degranulation [44]. Additionally, autophagy induction was shown to counter-regulate the receptor-interacting protein kinase 3 (RIPK3)-mixed lineage kinase-like (MLKL) signaling pathway required for cytolysis [23].

## 3. The Close Relationship between Degranulation and dsDNA Release in EET Formation

Eosinophils are potent cytotoxic effector cells that are able to release double-stranded deoxyribonucleic acid (dsDNA) in the extracellular space besides their capacity to discharge their granule content. In previous studies, the granule proteins were shown to colocalize with released mitochondrial DNA (mtDNA) to form so-called eosinophil extracellular traps (EETs) [2,45,46,47]. In our recent study, we demonstrated that mtDNA release from a bulk population of activated live eosinophils occurs subsequently to degranulation, implying different signaling pathways for these processes [4]. Considering their different release kinetics, it is likely that the association between eosinophil granule proteins and mtDNA during the formation of EETs occurs in the extracellular space [4].

EETs are formed upon eosinophil activation and have been demonstrated to occur in many different infectious diseases, allergic diseases and inflammatory skin diseases [48]. Whereas neutrophil extracellular trap (NET) formation is known to rely on glycolytic ATP production, active NADPH oxidase and rearrangement in the cytoskeleton [49,50], the exact molecular mechanism of EET formation remains unknown. Particularly, the origin of the DNA involved in EET formation is still a matter of debate [51,52]. EET formation has been reported to rely on a mtDNA scaffold from live cells [4,44,45,46,48,53,54] or respectively on a nuclear DNA scaffold from lytic cells undergoing eosinophil extracellular trap death (EETosis) [55,56,57,58,59,60]. *ATG5*-knockout eosinophils have been demonstrated with an increased ability to form EETs and consequently increased bacterial killing of *Escherichia coli* (*E. coli*) in vitro, as well as clearance of *Citrobacter rodentium* (*C. rodentium*) in vivo [44].

The released granule proteins MBP-1 [45,46], ECP [45,46,48] and EPX [4,44,54] were demonstrated to colocalize with the mtDNA scaffold. By contrast, no information is available regarding EDN and EET formation. Even though the exact role of the different granule proteins in EETs has not been determined, it is likely that the EETs can limit tissue damage by enabling a more focused action of the granule proteins in close proximity to a specific pathogen target.

## 4. Agonists and Methods Used in the Assessment of Eosinophil Degranulation

A variety of agonists (Table 1) and methods (Table 2) are used to assess the different types of eosinophil degranulation. Comparisons of the different studies are therefore difficult to perform or even impossible. In the following, we describe important agonists and methods to evaluate the degranulation capacity of eosinophils.

IL-33 is part of the IL-1 family members. This cytokine family comprises highly inflammatory cytokines, such as IL-1α, IL-1β and IL-18 [61]. The activation of the transcription factor NF-κB and the MAP kinases p38, JNK and ERK1/2 is characteristic for IL-1 receptor signaling [62]. The IL-33 precursor has been shown to contain a caspase-1 cleavage-site [63]. Cleavage of the IL-33 full-length protein by caspase-1 leads to the inactivation of the protein [64]. IL-33 acts through the IL-1 receptor ST2, a receptor shown to be expressed on eosinophils [65,66]. IL-33 was reported to be a potent eosinophil activator. Using different concentrations and incubation times, IL-33 was shown to induce the production of superoxide in eosinophils [66], to increase eosinophil adhesion [65,67] and to cause the release of different granule proteins such as EDN [66,67] and EPX [37,68] as well as ECP [68] (Table 2). Taken together, the proinflammatory effector functions of eosinophils can be stimulated by IL-33. However, the type of eosinophil degranulation induced by IL-33 has still not been defined (Table 1).

Interleukin-5 (IL-5) is a key biological factor involved in eosinophil differentiation, activation, survival and migration in tissues [69]. Eosinophils express the IL-5Rα subunit (CD125), to which IL-5 binds specifically. The IL-5R consists of IL-5Rα subunit and a nonspecific βc chain [70]. IL-5 was reported to cause PMD [71,72] (Table 1) and was shown to induce granule emptying [71,72], EPX release [72] and EDN release from eosinophils [66,67] (Table 2). EDN release and the loss of granule content were consistently higher following IL-33 stimulation, indicating IL-33 as a more potent agonist for EDN release than IL-5 [66,67,68].

Eotaxins are a subfamily of the CC chemokine family that acts on eosinophils via the CC chemokine receptor CCR3 [73,74]. The first discovered eotaxin was renamed to eotaxin-1 and later to CCL11 after eotaxin-2 and eotaxin-3 were found [75]; nevertheless, many publications still refer to it as eotaxin. Kampen et al. demonstrated a role of eotaxin in granule protein release and in chemotaxis, which were dependent on the rapid activation and phosphorylation of the MAP kinases ERK2 and p38 but not of JNK [76]. Eotaxin was shown to induce PMD [20]. Furthermore, the same group demonstrated that Brefeldin A, an inhibitor of vesicular transport, is able to inhibit eosinophil granule emptying following eotaxin stimulation [77]. Eotaxin was observed to induce granule emptying [20,39,78,79], IL-4 release [77], ECP release [76] and EDN release [80]. A recent study reported the lack of significant EPX release after eotaxin stimulation of IL-5- or IFN-γ-primed eosinophils [4]. Whether eotaxin is able to induce EPX release from eosinophil is not well defined and requires further investigations. These results are also influenced by agonist concentration and time periods of incubation, both of which have not been consistent among the studies.

TNF-α is a multifunctional cytokine able to promote cell proliferation, cytotoxicity, inflammation and immunomodulation [81]. Eosinophil apoptosis has been shown to be delayed by TNF-α acting via TNF-receptor 1. The resulting prolonged viability was dependent on the activation of NF-κB and AP-1 [82,83]. TNF-α has also been described to induce eosinophil granule discharging through compound exocytosis, a degranulation process characterized by the fusion of secretory granules [20,39,78,79]. In addition, an increase in adhesion as well as IL-8, GM-CSF and ECP release has been demonstrated following TNF-α stimulation [82]. Notably, in recent publications, the extent of granule protein release from TNF-α-activated eosinophils has been mostly assessed by granule density and degranulation pattern using TEM without targeting any of the eosinophil granule proteins.

PAF is a phospholipid that functions as a key chemoattractant for eosinophils [84,85,86,87]. This ligand acts by binding to the G-protein-coupled receptor PAFR [88,89], known to be expressed on eosinophils [90,91]. PAF was demonstrated to induce PMD using TEM [20]. Following PAF stimulation, eosinophils have been shown to release EPX [4,33,37,92,93], EDN [93] and ECP [14,94] (Table 2) mostly through PMD (Table 1). Likewise, Kim et al. demonstrated EDN release following activation with PAF [60]. However, in this study, PAF was used in a higher concentration and with increased incubation time compared to other publications. Importantly, cell toxicity was observed in response to the stimulation, a defining feature of cytolytic degranulation. It is therefore likely that the degranulation type induced by PAF is also dependent on the agonist concentration.

Additional agonists that are less commonly used (Table 1) have also been demonstrated to induce eosinophil degranulation. RANTES, such as PAF and eotaxin, has been described to cause PMD in eosinophils [20]. Eosinophils were shown to respond to *N*-formyl-methionyl-leucyl-phenylalanine (fMLP) by an increase in intracellular calcium, rate of oxygen consumption and the production of reactive oxygen species (ROS) [95]. Furthermore, fMLP is involved in human eosinophil degranulation [96] by induction of significant EDN release [67]. IFN-γ, a representative Th1 cytokine, was demonstrated to cause PMD in eosinophils [34,97]. In response to IFN-γ, RANTES release [97,98], EPX release [98] and EDN colocalization with CLC protein and CD63 occurred in eosinophils [93]. Moreover, IFN-γ-stimulation did not affect intracellular mobilization of MBP-1 [98]. Complement factor 5a (C5a) was shown to induce ECP release [94] and a maximum EPX release within one minute on cytokine-primed eosinophils [4]. Additionally, C5a alone did not induce an MBP-1 translocation [98]. Ionomycin, a calcium ionophore, has been shown to induce higher cytokine release compared to EDN release in eosinophils [99]. Furthermore, EPX release was demonstrated in response to ionomycin [4,33,37]. Another calcium ionophore, A23187, was shown to induce cytolysis in human eosinophils [55,60,100]. PMA, similar to fMLP, is involved in human eosinophil degranulation [96] and induces EPX release [4,41]. With longer incubation time (>1 h), PMA was demonstrated to cause cytolysis [55]. A combination of IVIG and iC3b has been shown to induce superoxide generation followed by early degranulation that finally leads to cytolysis. This process was dependent on the RIPK3-MLKL signaling pathway, which can be repressed by autophagy induction [23]. Eosinophils were demonstrated to undergo cytolysis upon exposure to the soluble plasma glycoprotein fibrinogen [101]. The adhesion to fibrinogen and its degradation by eosinophils have been shown to be CD11b-dependent [101]. Immobilized IgG and IgA were determined to cause both exocytosis [102] and cytolysis [55,102]. Finally, lysoPS is an endogenous lysophospholipid acting via P2Y10 receptor expressed by human eosinophils [103]. In a recent report, LysoPS was shown to induce EDN release [60]. Notably, cell toxicity is significantly increased with the LysoPS concentration used in this publication, suggesting cytolysis [60] (Table 1).

Typical methods employed to assess eosinophil degranulation comprise the measurement of granule protein concentrations by ELISA or the use of immunofluorescence and TEM (Table 2). CD63 is commonly stained as a surrogate marker to evaluate degranulation, either by measuring the surface upregulation by flow cytometry or by analyzing the ultrastructural distribution of CD63 by TEM in response to various stimuli. To date, TEM seems to be the most precise method to determine the ongoing degranulation process. The exact degranulation type induced by many agonists is still not well defined and needs further investigations (Table 1).

**Table 1 ijms-22-07091-t001:** Stimulation and agonists used to assess different modes of eosinophil degranulation.

Stimulation/Agonists	Degranulation Type	References
A23187	Cytolysis	[55,60,100]
C5a (+ IFN-γ * or IL-5 * or GM-CSF *^,+^)	---	[4] * [44] ^+^ [94] ^+^ [98]
CCL11 (Eotaxin-1) (+ IFN-γ * or IL-5 *^,+^)	Piecemeal degranulation (PMD)	[4] * [20,39,76] [77] ^+^ [78,79,80]
Fibrinogen	Cytolysis	[101]
IFN-γ	Piecemeal degranulation (PMD)	[34,93,97,98]
IL-33	---	[37,66,67,68,104]
IL-5	Piecemeal degranulation (PMD)	[66,67,68,71,72]
Immobilized IgG or IgA	Cytolysis	[55,102]
	Exocytosis	[102]
Ionomycin	---	[4,33,37]
IVIG + iC3b	Cytolysis	[23]
lysoPS	Cytolysis	[60]
fMLP	---	[67,96]
PAF (+ IL-5 * or IFN-γ * or IL-2 ^¥^ or GM-CSF ^£^)	Cytolysis	[60]
	Piecemeal degranulation (PMD)	[4] * [14] ^¥^ [20,33,37,92] [94] ^£^
	---	[93]
PMA	Cytolysis	[55]
	---	[4,41,96]
RANTES	Piecemeal degranulation (PMD)	[20]
TNF-α	Compound exocytosis	[20,39,78,79,82]

Each symbol (*, ^+^, ^¥^, ^£^) links a priming agent used before agonist stimulation to its reference in the same row.

**Table 2 ijms-22-07091-t002:** Methods and targeted proteins employed to assess eosinophil degranulation.

Measured/Targeted Protein	Method	References
Eosinophil Peroxidase (EPX)	Colorimetric assay	[4,41,92,98]
	Confocal laser scanning microscopy (CSLM) /Immunofluorescence staining	[4,68]
	ELISA	[33,37,38,93,97,105]
	Immunohistochemistry /Ultrastructural cytochemistry	[38,72]
	Mass spectrometry analysis	[101]
Eosinophil Cationic Protein (ECP)	Confocal laser scanning microscopy (CSLM) /Immunofluorescence staining	[68]
	Immunohistochemistry	[94]
	Pharmacia UniCap	[94]
	Radioimmunoassay	[76,82]
Eosinophil-derived Neurotoxin (EDN)	Confocal laser scanning microscopy (CSLM) /Immunofluorescence staining	[93]
	ELISA	[60,66,67,93,104]
	Radioimmunoassay	[80,100,102]
Major Basic Protein (MBP-1)	Confocal laser scanning microscopy (CSLM) /Immunofluorescence staining	[97,101]
	Dot blot assay	[37]
	Immunohistochemistry	[38]
	Immunonanogold electron microscopy (EM)	[106]
	Mass spectrometry analysis	[101]
Cell Surface Upregulation of Surrogate Degranulation Marker CD63	Flow cytometry (FACS)	[4,44,68,98,105]
CD63 Immunolabeling	Confocal laser scanning microscopy (CSLM) /Immunofluorescence staining	[33,37,98]
	Immunonanogold electron microscopy (EM)	[20,78,93]
Granzyme-B	Confocal laser scanning microscopy (CSLM) /Immunofluorescence staining	[68]
Charcot-Leyden Protein (CLC)	Confocal laser scanning microscopy (CSLM) /Immunofluorescence staining	[93,107]
	Immunonanogold electron microscopy (EM)	[71,72,107]
	Radioimmunoassay	[100]
IFN-γ	Immunonanogold electron microscopy (EM)	[79]
IL-4	ELISA	[77]
	Immunofluorescence staining	[77]
	Immunonanogold electron microscopy (EM)	[108]
Granule Density/ Ultrastructural analysis	Transmission electron microscopy (TEM)	[20,22,55,68,71,72,78,79,96,101,102,106]
	Phase-contrast microscopy	[55]
Qa-SNARE syntaxin-17 (STX17)	Immunonanogold electron microscopy (EM)	[39]
	Transmission electron microscopy (TEM)	[39]
RANTES (CCL5)	Confocal laser scanning microscopy (CSLM) /Immunofluorescence staining	[34,97,98]
	ELISA	[98]
	Immunonanogold electron microscopy (EM)	[97]

## 5. The Implication of Charcot–Leyden Crystal Protein in Degranulation

The eosinophil Charcot–Leyden crystal (CLC) protein, also known as galectin-10 (Gal-10), was discovered more than 150 years ago. CLC/Gal-10 is highly expressed in eosinophils and has the ability to form hexagonal bipyramidal CLC by autocrystallization [31]. It is a member of the Galectin superfamily and known to be a hallmark of eosinophil- and basophil-related inflammatory diseases [31].

CLC/Gal-10 has been shown to localize mainly in the peripheral cytosol of eosinophils [71,72,93,107]. Grozdanovic et al. demonstrated that CLC/Gal-10 interacts with EDN and ECP. Furthermore, they showed rapid colocalization of CLC/Gal-10 and EDN with CD63, a surrogate marker of degranulation, following IFN-γ stimulation [93]. In addition, CLC-/Gal-10-deficient eosinophils released significantly more EDN but not EPX following PAF stimulation [93]. These findings suggest that CLC/Gal-10 may function as a carrier for cationic ribonucleases and plays a role in the specific vesicular transport of these proteins during the piecemeal degranulation process. By contrast, Melo et al. could not demonstrate a colocalization between CLC/Gal-10 and the granule protein MPB-1 in resting eosinophils and did not observe a change in intracellular CLC localization following stimulation with CCL11 (agonist of PMD) or TNF-α (agonist of compound exocytosis) [107]. Moreover, CLC and EPX were shown to have different distributions in IL-5-stimulated eosinophils [72]. These data suggest a possible specificity of interaction of CLC with the eosinophil ribonucleases EDN and ECP.

Notably, the absence of CLC/Gal-10 significantly impaired the proliferation of eosinophil progenitors and granulogenesis leading to a reduced formation of specific granules, implying an important role of CLC/Gal-10 in eosinophil differentiation [93].

CLC formation has been recently demonstrated to be linked to eosinophil cytolysis [109,110]. The functional relevance of released CLC is still not fully understood and needs further investigations. In a recent research, Gevaert et al. tested the direct effect of CLC on different cell types [111]. CLC alone did not induce any change in the viability of epithelial cells. However, CLC-stimulated epithelial cells increased the migration and recruitment of neutrophils, confirming previous results [110,111,112]. Interestingly, GM-CSF-primed neutrophils stimulated with CLC demonstrated significant increase in NET formation, suggesting an important inflammatory role of CLC on neutrophils [111]. The exact mechanism triggered by CLC on neutrophils as well as a potential role of CLC in eosinophil cytolysis could not be confirmed in this study [111].

## 6. Function and Clinical Relevance of Eosinophil-Derived Granule Proteins, Cytokines and Chemokines

The main effector functions of eosinophils in major allergic and inflammatory diseases derive from the release of the granule content. Activated eosinophils secrete proinflammatory factors such as cytokines (IL-13, IL-5, osteopontin), chemokines (CCL11, CCL22, eotaxin), leukotrienes, matrix metalloproteinases and granule proteins [10,113]. These diverse mediators take action in destroying all types of microorganisms and in hypersensitivity reactions upon extracellular release following eosinophil activation [114]. Moreover, eosinophils have been suggested to help control tumor growth and metastasis formation in models of melanoma [115,116,117,118], colorectal carcinoma [119], fibrosarcoma [120], and hepatocellular and breast carcinoma [121,122].

The cationic granule proteins are able to stimulate other immune cells including neutrophils [123], basophils [124], mast cells [124] and dendritic cells [125], as well as nerve cells [126]. MBP-1 is stored in a nontoxic form in the crystalline core of the specific granules. Upon eosinophil activation, MBP-1 is cleaved and released as an active cytotoxic protein that exerts nonselective toxicity against bacteria, parasites and host tissue [127,128,129]. MBP-1 is found extracellularly in large amyloids consecutive to massive eosinophil infiltration and degranulation [129]. The presence of such MBP-1 deposition in the esophagus of symptomatic patients with eosinophilic esophagitis (EoE) suggest the use of extracellular MBP-1 as a marker of disease activity [130]. MBP-1 and ECP have been demonstrated to regulate mast cell functions including their activation and the release of diverse mediators such as histamine, PGD_2_, GM-CSF, TNF-α and IL-8 [131]. Furthermore, both proteins display antiparasitic and antibacterial properties [127,128,132].

A high level of ECP has been associated with several diseases including asthma [133,134], coronary artery disease [135] and several atopic and inflammatory diseases [133,134]. Moreover, increased ECP concentration is reported as a risk factor for ischemic stroke [136]. ECP and EDN promote production of ROS and the induction of apoptosis in keratinocytes [137]. The two proteins have been additionally demonstrated to upregulate matrix metalloproteinase-9 (MMP-9) expression [137]. Furthermore, EDN exhibits ribonuclease activity against single-stranded RNA viruses [138,139,140]. In addition, significant higher serum EDN levels were found in patients with eosinophilic chronic rhinosinusitis (ECRS). Thereby, EDN-induced secretion of MMP-9 results in airway remodeling and formation of intractable nasal polyps [141]. EPX exerts bactericidal [21,142], antiviral [143] and antiparasitic [144] activity by interacting with superoxide. A similar effect of the EPX-H_2_O_2_ system is observed in mammalian tumor cells [145] and on endothelial cells in eosinophilic endocarditis [146].

In EoE patients, baseline EPX concentration in tissue biopsies is found to be significantly increased [147]. Conversely, no significant difference in absolute level of EPX, EDN or ECP were observed in the serum of EoE patients, while EPX and EDN serum proteins levels normalized for AEC were significantly decreased [148]. Similarly, no evidence of increased degranulation or morphologic granule difference of circulating eosinophils has been detected in other allergic diseases including asthma, atopic dermatitis, allergic rhinitis and eosinophilic granulomatosis with polyangiitis (EGPA, formerly Churg-Strauss syndrome) compared to healthy individuals [149]. Elevated ECP, EPX, EDN and MBP-1 plasma levels together with an increased AEC have been described in many helminth-infected patients [150]. Additionally, MBP-1, ECP and EPX have been found in association with mtDNA in the formation of EETs [4,45,46,48,151]. These extracellular structures are observed in infectious, allergic and autoimmune eosinophilic diseases [2]. Serum EDN concentrations and counts of EET-forming eosinophils are increased in severe asthma compared to nonsevere asthma, suggesting a role of eosinophil degranulation and EET formation in type-2 airway inflammation [60].

CLC/Gal-10 has been found at sites of infection mediated by helminths [152], fungi [153,154] and bacteria [155]. Interestingly, CLC/Gal-10 has been reported to be downregulated at mRNA gene expression level in patients with respiratory syncytial virus (RSV) infection [156]. Furthermore, CLC/Gal-10 has been detected in celiac disease [157], asthma and other allergic diseases [158,159,160], as well as in hypereosinophilic syndrome (HES) [161]. In the sputum of asthmatic patients, the expression of CLC/Gal-10 allowed for the discrimination between inflammatory phenotypes [162] and strongly correlated with the number of eosinophils [163].

A number of various cytokines, chemokines and growth factors are produced by eosinophils, some of them being stored as preformed mediators in specific granule as well as in small secretory vesicles, including CCL5/RANTES, CCL11/eotaxin, GM-CSF, IL-2, IL-4, IL-5, IL-6, IL-13, TGF-α and TNF-α [9]. Their expression and secretion have been reported in both peripheral blood and tissue eosinophils. The released cytokines are believed to amplify and regulate localized immune responses [164]. For instance, eosinophil-derived inflammatory mediators seem to be involved in several eosinophilic diseases such as allergic rhinitis [9], atopic and nonatopic asthmatics [165], celiac disease [166], eosinophilic cystitis [167], hypereosinophilic syndrome [167] and eosinophilic heart disease [168]. A network of cytokines and chemokines has been shown to orchestrate inflammatory processes of allergic reactions through the regulation of IgE responses, bone marrow progenitor cell differentiation and the expression of adhesion molecules [9]. Specifically, several Th2 cytokines, including IL-4 and IL-5, are known to be involved in eosinophilic inflammatory diseases especially asthma [169]. IL-5 and eotaxin-3 have been found to work synergistically in chronic subdural hematomas (CSDH). The subsequent infiltration of eosinophils and induction of EDN degranulation have been shown to lead to the growth of CSDH [170]. IL-33 was reported to inhibit tumor growth in colorectal cancer [171] and skin cancer [118] through a mechanism involving eosinophil recruitment, activation and degranulation. Moreover, RNAi silencing of chemokine receptor 3 (CCR3) inhibited eosinophil degranulation resulting in reduced inflammation in allergic rhinitis [172,173].

Lysophosphatidylcholines (LPCs) have been reported to upregulate CD11b surface expression and to modulate eosinophil effector functions such as degranulation, chemotaxis and downstream signaling through the disruption of cholesterol-rich nanodomains on cell membranes. LPCs are generated by phospholipase that are released by eosinophils upon allergen exposures [174].

## 7. Concluding Remarks

Eosinophils are granule-containing white blood cells that are recently discovered to be involved in immunoregulatory and homeostatic functions besides their traditional role as cytotoxic effector cells. Degranulation is one of the key effector functions of these cells and has been extensively studied. Eosinophils have been associated with numerous allergic and nonallergic inflammatory diseases through the presence of granule proteins at the site of inflammation or infection. Herein, we highlight the fact that, despite extensive studies and a good knowledge of the ultrastructural pattern of the four major degranulation mechanisms occurring in eosinophils, little is known regarding their respective molecular modulation. Additionally, only a few agonists (Figure 1) have been well characterized for a defined type of degranulation considering the huge variety of stimuli reported in the literature. The variances in concentration and incubation time of a specific agonist complicate the comparison between studies and draw attention to the lack of uniformity in the field. Enhanced understanding of the role of degranulation in health and disease requires additional experimentations to define the type of degranulation induced by the different agonists. Furthermore, the improvement of our knowledge regarding the complex mediator release specificity in response to different stimulations will lead to a better understanding of eosinophil-related pathologies.

## Figures and Tables

**Figure 1 ijms-22-07091-f001:**
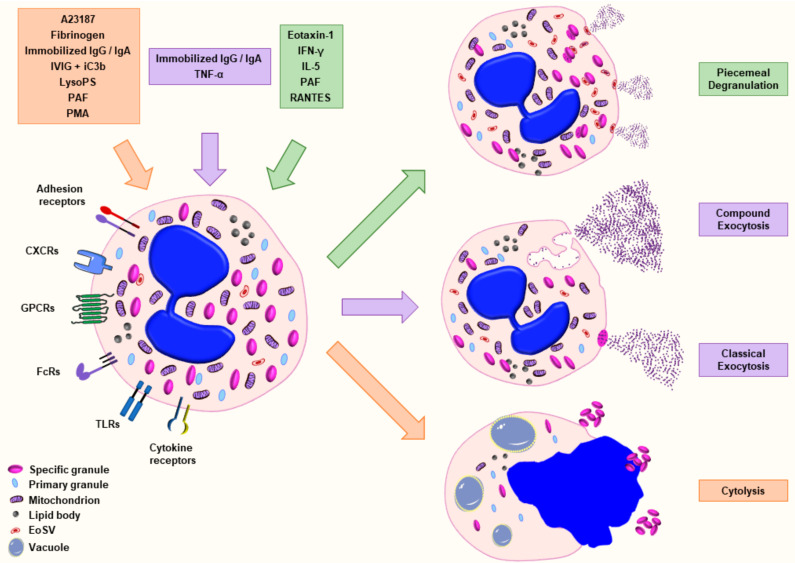
Schematic representation of the major mechanisms of eosinophil degranulation. Eosinophils are able to release granule content by piecemeal degranulation (PMD), classical exocytosis, compound exocytosis or cytolysis in response to various stimuli. Piecemeal degranulation describes the progressive and selective release of granule content mediated by transport vesicles. Classical exocytosis defines the release of the entire granule content by fusion of the granule with the plasma membrane. Compound exocytosis is another type of exocytosis characterized by intracellular granule–granule fusion prior to extracellular release. Cytolysis indicates a nonapoptotic form of cell death defined by the formation of vacuoles within the cells, the disintegration of nuclear and plasma membrane leading to the release of nuclear DNA and the deposition of intact granules in the outer space. The agonists that have been well characterized for a defined type of degranulation are illustrated with the corresponding color. Abbreviations: EoSV, eosinophil sombrero vesicle; iC3b, inactive complement component 3b; IFN-γ, interferon-γ; IgA, immunoglobulin A; IgG, immunoglobulin G; IL, interleukin; IVIG, intravenous immunoglobulin; LysoPS, lysophosphatidylserine; PAF, platelet-activating factor; PMD, piecemeal degranulation; and TNF-α, tumor necrosis factor-α.

**Figure 2 ijms-22-07091-f002:**
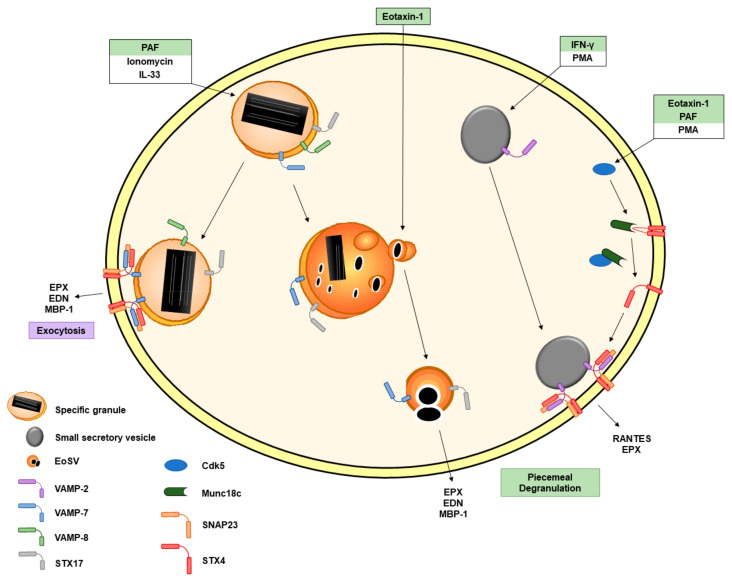
Schematic representation of the involvement of SNARE proteins in eosinophil degranulation. SNARE proteins are intracellularly present in eosinophils. VAMP-8 is found on specific granules. VAMP-7 is also expressed on specific granules and eoSVs. PAF, ionomycin and IL-33 are described to be involved in VAMP-7-mediated eosinophil granule protein release. Syntaxin-17 is present in specific granules and eoSVs. Eotaxin-1 leads to a significant increase in syntaxin-17^+^ secretory vesicles. VAMP-2 is expressed on secretory vesicles. IFN-γ induces translocation of VAMP-2^+^ vesicles to the plasma membrane where VAMP-2 interacts with SNAP23 and STX4. The inhibitory binding of Munc18c to STX4 is disrupted by the association of Cdk5 with Munc18c following eosinophil activation with eotaxin-1, PAF and PMA. The agonists are labelled with the color corresponding to the type of degranulation illustrated in Figure 1. Abbreviations: Cdk5, cyclin-dependent kinase 5; ECP, eosinophil cationic protein; EDN, eosinophil-derived neurotoxin; EoSV, eosinophil sombrero vesicle; EPX, eosinophil peroxidase; IFN-γ, interferon-γ; IL, interleukin; PAF, platelet-activating factor; PMA, phorbol 12-myristate 13-acetate; PMD, piecemeal degranulation; STX4, syntaxin-4; STX17, syntaxin-17; TNF-α, tumor necrosis factor-α; and VAMP, vesicle-associated membrane protein.

## Data Availability

Not applicable.

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
