# Peer review of "The Enigma of Eosinophil Degranulation"

_ijms, 2021, doi:10.3390/ijms22137091_

Round 1
Reviewer 1 Report
Congratulations to Fetterelt et al. - this is a very nice and interesting piece.
I think this is an important contribution to and resource for the field and will be much cited.
Only few minor edits:
Line 15: I would suggest to omit the word "intestinal" as regulatory eosinophils seem to play an important role in lung homeostasis as well
Line 29: i think evolutionally should be changed to evolutionary
Line 41: Please try to make it more clear that the "specific granules" are the 2nd type of granules in addition to the "primary granules", for instance by starting the sentence with "In contrast, the specific granules are formed..."; Another option would be to format "primary granules" and "specific granules" (throughout the text) in italics.
Line 46: Please add a reference after the end of the sentence
Line 46: I suggest to re-structure the sentence to: "They seem to have immunoregulatory functions and are involved in..."
Fig legends: I think it might be helpful to state the abbreviations also directly in the figure legends, not only at the end of the text.
line 117: do the authors mean Q-SNARE (rather than Qa-SNARE)?
lines 167: i think it might be interesting to briefly also discuss what is known about molecular mechnisms of neutrophil extracellular traps as this could be a similar situation in eosinophils
line 278: i think it thould be "has" rather than "have"
line 389: is CD11b upregulation an effector function of eosinophils? please re-phrase the sentence if not.
Author Response
Congratulations to Fetterelt et al. - this is a very nice and interesting piece.
I think this is an important contribution to and resource for the field and will be much cited.
Reply: We thank the reviewer for the appreciation of our work.
Only few minor edits:
Line 15: I would suggest to omit the word "intestinal" as regulatory eosinophils seem to play an important role in lung homeostasis as well
Reply: The word “intestinal” has been removed from the sentence (line 16)
Line 29: i think evolutionally should be changed to evolutionary
Reply: The word “evolutionally” has been replaced by “evolutionary” (line 30)
Line 41: Please try to make it more clear that the "specific granules" are the 2nd type of granules in addition to the "primary granules", for instance by starting the sentence with "In contrast, the specific granules are formed..."; Another option would be to format "primary granules" and "specific granules" (throughout the text) in italics.
Reply: To avoid any confusion, we rephrase the sentence as follows: “The specific granules, also termed crystalloid or secondary granules, are formed during the myelocyte stage and are characterized by their crystalline core [6].” (line 43-45)
For more consistency, we used the term “specific granules” throughout the text.
Line 46: Please add a reference after the end of the sentence
Line 46: I suggest to re-structure the sentence to: "They seem to have immunoregulatory functions and are involved in..."
Reply: The reference is the same as for the next sentence. To avoid any confusion, we replaced “They” with “Lipid bodies” and added the reference after the 1st sentence (lines 49)
Fig legends: I think it might be helpful to state the abbreviations also directly in the figure legends, not only at the end of the text.
Reply: The abbreviations have been added in the figure legends (Fig. 1, lines 86-89; Fig. 2, lines 161-164)
line 117: do the authors mean Q-SNARE (rather than Qa-SNARE)?
Reply: Qa-SNARE is a subtype of Q-SNARE. We used the term “Qa-SNARE syntaxin-17” as it was used by Carmo et al. 2015 (line 124)
lines 167: i think it might be interesting to briefly also discuss what is known about molecular mechnisms of neutrophil extracellular traps as this could be a similar situation in eosinophils
Reply: We added a sentence regarding the molecular mechanisms involved in NET formation as follow: “Whereas neutrophil extracellular trap (NET) formation is known to rely on glycolytic ATP production, active NADPH oxidase and rearrangement in the cytoskeleton [49,50], the exact molecular mechanism of EET formation remains unknown.” (lines 178-180)
line 278: i think it thould be "has" rather than "have"
Reply: We replaced “have” with “has” (line 297)
line 389: is CD11b upregulation an effector function of eosinophils? please re-phrase the sentence if not.
Reply: We re-phrased the sentence as follow: “Lysophosphatidylcholines (LPCs) have been reported to upregulate CD11b surface expression and to modulate eosinophil effector functions such as degranulation, chemotaxis and downstream signaling through the disruption of cholesterol-rich nanodomains on cell membranes.” (lines 407-410)
Reviewer 2 Report
I think this is a useful review for researchers in this field, but I think it would be more valuable as a review if table 1 was more comprehensive.
It has been shown that other stimuli also induce cytolysis (see recent review by Fukuchi et al. Table 1. DOI: 10.1016/j.alit.2020.10.002).
In line 39-40, the authors stated that CLC-P (galectin-10) is in primary granules but stated in the peripheral cytosol (line 282). The “primary” and “secondary” granules may be outdated descriptions (see review by Melo and Weller. DOI: 10.1002/JLB.3MR1217-476R).
Author Response
I think this is a useful review for researchers in this field, but I think it would be more valuable as a review if table 1 was more comprehensive.
Reply: We thank the reviewer for the appreciation of our work. We have now expanded the Table 1 (Page 8, line 292) and added additional references.
It has been shown that other stimuli also induce cytolysis (see recent review by Fukuchi et al. Table 1. DOI: 10.1016/j.alit.2020.10.002).
Reply: The different stimuli that were reported to induce “cytolysis” are now added to the “Table 1” with the related references. The “Table 2” (Page 9-10, line 294) and the text (Lines 269-279) have been updated accordingly.
In line 39-40, the authors stated that CLC-P (galectin-10) is in primary granules but stated in the peripheral cytosol (line 282). The “primary” and “secondary” granules may be outdated descriptions (see review by Melo and Weller. DOI: 10.1002/JLB.3MR1217-476R).
Reply: To avoid any confusion, we included the following sentence: “In this context, it should be mentioned that primary granules were recently suggested to be early “specific” cored granules [7].” (Lines 42-43).